# Reducing Uncertainty in 3D Medical Image Segmentation under Limited Annotations through Contrastive Learning

**Sanaz Karimijafarbigloo** [1]                    SANAZ.KARIMIJAFARBIGLOO@UR.DE
**Reza Azad** [2]                                          AZAD@PC.RWTH-AACHEN.DE
**Amirhossein Kazerouni** [3]                        AMIRHOSSEIN477@GMAIL.COM
**Dorit Merhof** [1,4]                    DORIT.MERHOF@INFORMATIK.UNI-REGENSBURG.DE

[1] *Faculty of Informatics and Data Science, University of Regensburg, Regensburg, Germany*

[2] *RWTH Aachen University, Aachen, Germany*

[3] *School of Electrical Engineering, Iran University of Science and Technology, Tehran, Iran*

[4] *Fraunhofer Institute for Digital Medicine MEVIS, Bremen, Germany*

**Editors:** Accepted for publication at MIDL 2024

## Abstract

Despite recent successes in semi-supervised learning for natural image segmentation, applying these methods to medical images presents challenges in obtaining discriminative representations from limited annotations. While contrastive learning frameworks excel in similarity measures for classification, their transferability to precise pixel-level segmentation in medical images is hindered, particularly when confronted with inherent prediction uncertainty. To overcome this issue, our approach incorporates two subnetworks to rectify erroneous predictions. The first network identifies uncertain predictions, generating an uncertainty attention map. The second network employs an uncertainty-aware descriptor to refine the representation of uncertain regions, enhancing the accuracy of predictions. Additionally, to adaptively recalibrate the representation of uncertain candidates, we define class prototypes based on reliable predictions. We then aim to minimize the discrepancy between class prototypes and uncertain predictions through a deep contrastive learning strategy. Our experimental results on organ segmentation from clinical MRI and CT scans demonstrate the effectiveness of our approach compared to state-of-the-art methods. Code.

**Keywords:** Uncertainty, Contrastive, Segmentation, Medical Image.

## 1. Introduction

Medical image segmentation plays a pivotal role in the field of medical imaging, serving as a crucial step in the analysis and interpretation of complex visual data. This process involves the partitioning of images into meaningful and clinically relevant regions, allowing for a detailed examination of structures and abnormalities within the human body. One prominent approach to medical image segmentation is supervised learning. This paradigm involves training algorithms on labeled datasets, where each image is accompanied by annotations identifying the regions of interest. Despite its success in many applications, its main drawback lies in its dependency on large and accurate labeled datasets (Aljuaid and Anwar, 2022; Azad et al., 2023). Creating such datasets for medical images requires considerable expertise and time. Additionally, supervised learning is susceptible to human error caused by manual segmentation and labeling.

In response to these challenges, various strategies have been proposed to alleviate the dependency on meticulous labeling processes. Unsupervised learning operates without labeled

data. Algorithms in this category identify inherent patterns and structures within medical images without prior knowledge of specific regions (Caron et al., 2018; Chen and Frey, 2020; Hamilton et al., 2022; Zhao et al., 2022; Feng et al., 2023; Omidi et al., 2024). However, its limitations include the difficulty of distinguishing between normal and abnormal structures, hindering applicability in clinical settings that require precise identification. Interpretability of results is also challenging, as decisions rely solely on inherent data patterns. Transfer learning is another powerful strategy for enhancing medical image segmentation, leveraging pre-trained models to improve performance on tasks with limited labeled data (Kora et al., 2022; Araújo et al., 2022; Alhares et al., 2023). However, its drawback stems from the assumption of similar distributions between the source and target domains. If this assumption is not met, pre-trained features may not capture the nuances of the target medical imaging data, potentially leading to suboptimal results. Self-supervised learning overcomes label scarcity by generating its own supervisory signals, enhancing model performance (Tang et al., 2022; Karimijafarbigloo et al., 2023; Ouyang et al., 2022; Kazerouni et al., 2023). This approach, beneficial when labeled data is limited, faces the challenge of designing effective surrogate tasks. Ensuring these tasks capture pertinent information is crucial for the success of self-supervised methods. Furthermore, these approaches usually require additional supervisory signals derived from annotated data to be specifically directed toward the targeted task.

To address these issues, semi-supervised learning (SSL) offers a promising solution. This approach involves training models with a limited number of labeled samples and a large number of unlabeled data, striking a balance between supervised and unsupervised methods. In medical image segmentation, it proves valuable in scenarios where obtaining a fully labeled dataset is impractical, providing a practical and cost-effective solution for training robust segmentation models (Luo et al., 2022b; Wu et al., 2022; Luo et al., 2022a; Wang et al., 2023). SSL commonly employs two primary methods: pseudo-labeling and consistency regularization. Pseudo labeling involves generating "pseudo labels" for unlabeled data using a model's predictions. In a teacher-student (Tarvainen and Valpola, 2017; Wang et al., 2022; Xu et al., 2021) scenario, the teacher model, represented as the EMA of the student model, plays the role of pseudo labels generation. These pseudo labels are then integrated with the original labeled dataset to optimize accuracy and achieve cost-effective training (Lee et al., 2013; Xie et al., 2020). However, it's important to acknowledge that while the pseudo-labeling approach offers advantages, the scarcity of labeled data raises concerns about the reliability of pseudo-label quality. Hence, this method may introduce the risk of inaccuracies in the training data, potentially affecting the final model's precision.

To address the mentioned issue, current methods suggest adopting confidence score filtering for predictions (Zuo et al., 2021; Zou et al., 2021; Zhang et al., 2021; Sohn et al., 2020). This means that only the predictions with high confidence scores are employed as pseudo-labels, while those that are uncertain are disregarded. Nevertheless, this approach is not perfect in removing inaccurate predictions, as some incorrect predictions might possess high classification scores, known as over-confidence or miscalibration (Guo et al., 2017). Furthermore, setting a high threshold would significantly decrease the quantity of generated pseudo-labels, thereby restricting the efficacy of semi-supervised learning. Additionally, the potential problem of solely relying on presumably reliable predictions (which may instead comprise inadequate representations of certain classes or segments) may lead to an imbal-

ance of the training data and ultimately compromise the model's performance, particularly for challenging and less frequent classes. For instance, when the model encounters difficulty in accurately predicting specific classes, generating accurate pseudo-labels for the corresponding pixels becomes problematic. In this respect, Lu et al. (Lu et al., 2023) suggested a technique that combines pseudo-labeling with dual consistency regularization, emphasizing its strong uncertainty awareness capability. This approach incorporates a cycle-loss regularization to enhance the accuracy of uncertainty estimation. Shen et al. (Shen et al., 2023) introduced the UCMT method for semi-supervised semantic segmentation. This approach consists of two main components: Collaborative Mean-Teacher (CMT) and an uncertainty-guided region mix. The CMT component aims to maintain model disagreement while enhancing the quality of pseudo-labels through collaboration. Specifically, UCMT generates a new image by replacing uncertain regions with certain ones and then utilizes a collaborative approach to ensure consistent predictions across different networks. However, a limitation of this method is that it does not explicitly modify the representation of related voxels to reduce uncertainty. Consequently, there is a need for a mechanism to re-represent these uncertain voxels with different localities, such as through deformable convolutions, which could significantly enhance the overall effectiveness of the approach.

Acknowledging the reliability concerns linked with pseudo-labeling and the drawbacks of confidence score filtering, our method introduces a novel semi-supervised contrastive learning approach to address these challenges. In this context, we outline the following key contributions: **1)** We propose a mechanism to recognize uncertain predictions as a means to refine network representation, aiming for improved overall representation quality. **2)** To alleviate prediction uncertainty, we introduce an uncertainty-aware feature descriptor module. This module enhances contextual and semantic representation, contributing to a more robust and accurate prediction. **3)** We design a deep contrastive supervision function to minimize discrepancies between class prototypes and uncertain predictions.

## 2. Method

Current SSL algorithms that rely on consistency learning, such as Mean-teacher (Tarvainen and Valpola, 2017) and (Chen et al., 2021), propose applying consistency regularization not within a single model but among the pseudo labels within a multi-model architecture. Nonetheless, throughout the training process, there is a tendency for the dual-network SSL framework to quickly reach a consensus, causing the co-training to degrade into self-training (Kendall and Gal, 2017). To address this issue, we introduce a setup consisting of a predictive model accompanied by an auxiliary model. The predictive model serves as a pseudo-label generator, directing the training of the other model. In our strategy, we work with two distinct datasets: $\mathcal{D}_l$, comprising labeled data pairs $(\mathbf{x}_i^l, \mathbf{y}_i^l)$, where $i$ ranges from 1 to $N_l$, and $\mathcal{D}_u$, a considerably larger unlabeled dataset denoted as $\{x_i^u\}_{i=1}^{N_u}$, $N_l$ and $N_u$ indicate the number of samples in the labelled and unlabled dataset, respectively. The goal is to synergistically leverage the inherent potential of the unlabeled data to enhance feature representation, effectively training our semantic segmentation model by integrating a limited set of labeled data with an extensive repository of unlabeled data. For labeled data, both predictive and auxiliary models undergo optimization through supervised learning:

$$\mathcal{L}_s = \frac{1}{|\mathcal{B}_l|} \sum_{(\mathbf{x}_i^l, \mathbf{y}_i^l) \in \mathcal{B}_l} \ell_{ce}(\hat{\mathbf{y}}_i^l, \mathbf{y}_i^l) + \text{Dice}(\hat{\mathbf{y}}_i^l, \mathbf{y}_i^l) + \ell_{ce}(\tilde{\mathbf{y}}_i^l, \mathbf{y}_i^l) + \text{Dice}(\tilde{\mathbf{y}}_i^l, \mathbf{y}_i^l) \tag{1}$$

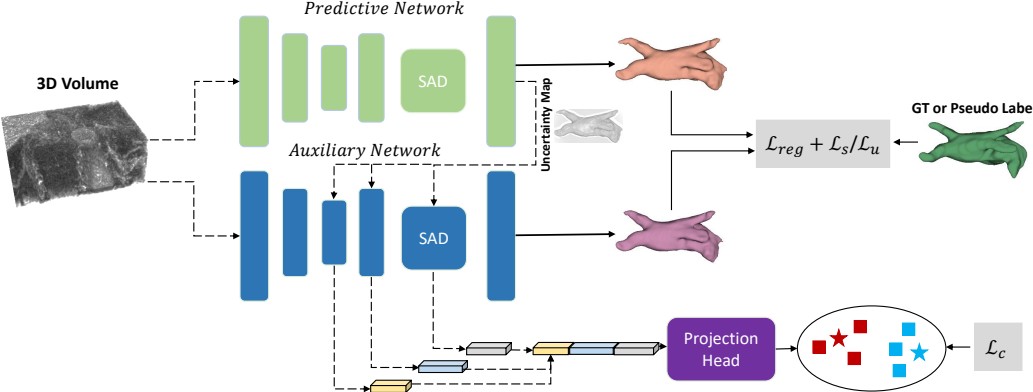

Figure 1: The dual-path semi-supervised contrastive learning method proposed aims to harness unsupervised data while minimizing uncertain predictions.

where $\hat{\mathbf{y}}_i^l$, $\tilde{\mathbf{y}}_i^l$ denotes the predicted segmentation map of the predictive and auxiliary networks, respectively. $B_l$ indicates the batch of labeled data. When handling unlabeled data, two key factors come into play: 1) direct guidance from the predictive model to the auxiliary model to reduce uncertainty and 2) minimizing uncertain predictions through the definition of deep-supervisory contrastive learning. To accomplish the former, we first fed the unlabeled sample into the predictive model to generate the pseudo label $\hat{\mathbf{y}}_i^u$. Then using the pseudo label we calculate the prediction loss for the auxiliary models. To further enhance error correction, we incorporate a regularization loss $\mathcal{L}_{reg}$, which calculates the $L_1$ distance between the XOR predictions (Wang et al., 2023) of the predictive and the auxiliary models:

$$\mathcal{L}_u = \frac{1}{|\mathcal{B}_u|} \sum_{\mathbf{x}_i^u \in \mathcal{B}_u} \ell_{ce}(\tilde{\mathbf{y}}_i^u, \hat{\mathbf{y}}_i^u) + \text{Dice}(\tilde{\mathbf{y}}_i^u, \hat{\mathbf{y}}_i^u) + \lambda_{reg}\mathcal{L}_{reg}(\tilde{\mathbf{y}}_i^u, \hat{\mathbf{y}}_i^u), \qquad (2)$$

where $\lambda_{reg}$ indicates the weight of the $\mathcal{L}_{reg}$. We additionally restrict the computation of the loss function to voxels with prediction confidence exceeding 0.7. The overall training loss ($\mathcal{L}$) incorporates three components: the supervised loss ($\mathcal{L}_s$), unsupervised loss ($\mathcal{L}_u$), and the contrastive loss ($\mathcal{L}_c$). Details of the contrastive loss are presented in subsection 2.2. The optimization objective revolves around minimizing the overall loss, expressed as:

$$\mathcal{L} = \mathcal{L}_s + \lambda_u\mathcal{L}_u + \lambda_c\mathcal{L}_c, \qquad (3)$$

The weights $\lambda_u$ and $\lambda_c$ determine the contribution of the unsupervised and contrastive losses, respectively. The overall architecture of the network is depicted in Figure 1.

## 2.1. Multi-Branch Contextual Uncertainty Reduction (MultiCURE) Module

In our design, we argue that inaccuracies can arise due to inadequate contextual information in the vicinity of voxels, which makes precise predictions challenging. To address this issue, we first input the sample into the predictive model to generate the segmentation map. Then, We create a binary uncertainty map by thresholding the softmax output of the network. In this process, pixels with confidence scores below a specified threshold are labeled as uncertain regions (assigned a value of 1), while pixels with confidence scores above the

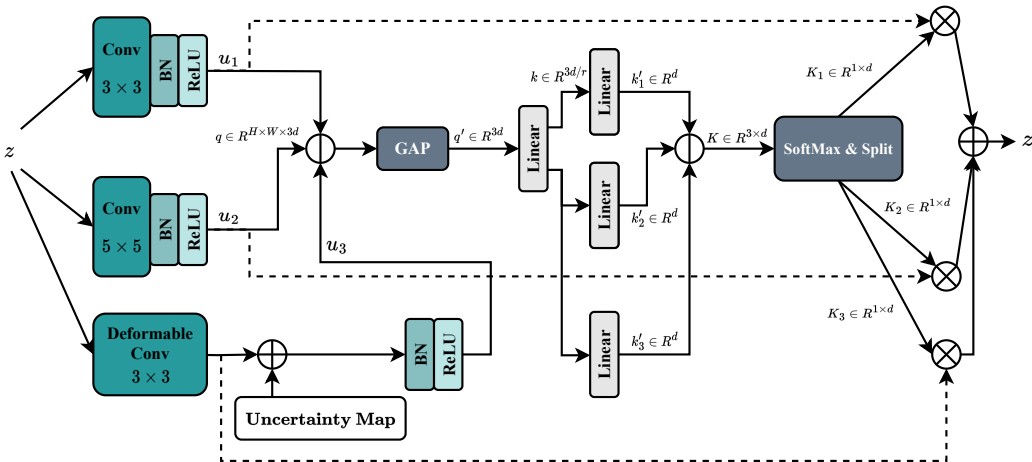

Figure 2: **MultiCURE Module:** Adaptive feature recalibration across scales using three branches, overcoming fixed kernel limitations with deformable convolutions, selectively extracting information from uncertain and certain regions, and integrating through channel-wise concatenation with a soft attention mechanism.

threshold are labeled as certain regions (assigned a value of 0). Finally, we enhance the representation of uncertain voxels on the auxiliary map using the MultiCURE module.

MultiCURE, a key component in our proposed architecture, is designed to recalibrate context information across multiple scales by adaptively selecting receptive fields from global and local pathways. It effectively overcomes the limitations of fixed kernel sizes in traditional convolutions, reducing uncertainty in critical areas, especially along object boundaries, which inherently have the highest degree of uncertainty. Comprising three branches, two maintain fixed kernel sizes, enhancing the flexibility of receptive field sizes. However, grid misalignment issues persist, particularly along object boundaries (or uncertain regions), resulting in less optimal results. The third branch provides vital global information, refining boundary delineation, and significantly helps to improve overall performance.

The MultiCURE employs a three-path split, incorporating convolution, Batch Normalization (BN), and ReLU activation function in two paths with $3 \times 3$ and $5 \times 5$ kernel sizes. The third path performs a $3 \times 3$ Deformable convolution (Dai et al., 2017) on the input $z \in R^{H \times W \times d}$. The deformable convolution in our design dynamically adjusts the receptive field for each feature map location by using an offset field to flexibly warp the sampling grid. This allows the model to handle varied object sizes in an image and gain a superior understanding of object regions. To further enhance feature representation within uncertain regions, we integrate (i.e., sum) an uncertain attention map with the output of the deformable convolution. This integration conditions the representation based on uncertain regions, serving as a mechanism to accentuate the representation of these areas using non-fixed resampling points. Consequently, this approach allows us to allocate increased attention to uncertain regions.

To integrate information from all branches, a fusion step entails performing channel-wise concatenation: $q = [u_1||u_2||u_3] \in R^{H \times W \times 3d}$. The next step involves applying global average pooling (GAP) to $q$ to condense the spatial information across the entire feature

map. In this context, the vector $q' \in \mathbb{R}^{3d}$ undergoes a linear transformation denoted as $\mathcal{F} : q' \to k$, resulting in the vector $k$ in $\mathbb{R}^{3d/r}$. This transformation serves the purpose of dimensionality reduction, thereby enhancing computational efficiency. Subsequently, each path follows an independent linear layer to revert the transformed vector of dimension $3d/r$ back to its original dimensionality, denoted as $d$ in the original input.

By concatenating all paths within the channel dimension ($K \in R^{3 \times d}$), a soft attention mechanism, specifically the SoftMax function, is deployed across channels. This adaptive approach facilitates the selective emphasis on the most pertinent feature scales. The resultant feature map, denoted as $z' \in R^{H \times W \times d}$, is attained by applying attention weights to kernels associated with the individual streamlines: $z' = u_1 \cdot K_1 + u_2 \cdot K_2 + u_3 \cdot K_3$

## 2.2. Deep Contrastive Learning

In our methodology, we propose deep contrastive supervision to refine the model's representation and enhance its discriminative capabilities. Initially, we identify high-confidence and uncertain predictions within the segmentation output. Leveraging the high-confidence predictions, we define class prototypes by extracting representations from multiple network levels, to capture both shallow and depth representation. To compute each prototype, we calculate the mean vector ($c$) of reliable voxel representations for class $k$ by defining $c_k$ as:

$$\mathbf{c}_k = \frac{1}{|S_k|} \sum_{(\mathbf{v^r}_i, y_i) \in S_k} f_{l:L}(\mathbf{v^r}_i), \tag{4}$$

where $f_{l:L}(\mathbf{v^r}_i)$ represents the feature representation of the voxel corresponding to the reliable predictions $(\mathbf{v^r}_i, y_i)$ from different levels of the network and $S_k$ is the set of certain predictions for class $k$. Subsequently, for the set of uncertain predictions $f(\mathbf{v^u r}_i)$, we resample candidates to align them with the corresponding class prototype, employing a contrastive learning algorithm for this purpose. By applying contrastive learning to feature sets extracted from various network blocks, we provide a deep supervisory signal for the network to contextually recalibrate the representation of uncertain pixels, aligning them with the class prototype. The contrastive loss is computed by aggregating their representations, and our objective is to minimize this loss. This approach leverages multi-level representations for deep supervision, enabling the refinement and improvement of the network's predictions to generate more discriminative features.

$$\mathcal{L}_{c_k} = -\frac{1}{|S_k|} \sum_{(\mathbf{v^u}_i, y_i) \in S_k} \log \frac{\exp\left(\frac{\text{sim}(\mathbf{v^u}_i, \mathbf{c}_k)}{\tau}\right)}{\sum_{j \neq k} \exp\left(\frac{\text{sim}(\mathbf{v^u}_i, \mathbf{c}_j)}{\tau}\right)}, \tag{5}$$

where, $\mathcal{L}_{c_k}$ is the contrastive loss for class $k$, sim $(\mathbf{v^u}_i, \mathbf{c}_k)$ is the similarity function measuring the cosine similarity between an uncertain voxel representation $\mathbf{v^r}_i$ and the class prototype $\mathbf{c}_k$, $\tau$ is a temperature parameter controlling the sharpness of the contrastive loss function. We augment the contrastive loss by adding an additional term that considers the distance between class prototypes. This extra term is incorporated to promote the representation space to actively separate the clustering spaces of different classes.

## 3. Experiments

We implemented a two-stream pipeline in PyTorch, utilizing an RTX A5000 GPU. Following (Wang et al., 2023), Vnet and Resnet were chosen for the predictive and auxiliary models, respectively. The training involved 6000 iterations with a batch size of 2, sampling randomly from supervised and unsupervised sources. SGD optimizer with parameters: decay factor 0.0001, momentum 0.9, and initial learning rate 0.01 (decayed by a factor of 10 every 2500 iterations). To manage hyperparameters, we set $\lambda_u = 1.0$ and $\lambda_c = 0.1 * e^{4(1-t/t_{max})^2}$ for dynamic weighting, where $t$ and $t_{max}$ denote current and maximum iterations. For evaluation, we follow (Wang et al., 2023) setting and use 5-fold and 4-fold cross-validation on the LA and Pancreas datasets, respectively.

### 3.1. Dataset

**Left Atrial Dataset (LA):** This dataset (Xiong et al., 2021) consists of 100 3D gadolinium-enhanced MR imaging volumes with non-uniform resolution ($0.625 \times 0.625 \times 0.625$ mm$^3$) and manual annotations for the left atrial region. Following the pre-processing protocol from (Wang et al., 2023), we normalized volumes to zero mean and unit variance. During training, random cropping used with dimensions of $112 \times 112 \times 80$. For inference, we used a sliding window approach ($112 \times 112 \times 80$) with a stride of $18 \times 18 \times 4$.

**NIH Pancreas Dataset:** This dataset (Roth et al., 2015) consists of 82 abdominal CT volumes annotated for the pancreas. We preprocess it by applying soft tissue windowing (HU range: -120 to 240) and spatial alignment using a method from (Luo et al., 2021; Wang et al., 2023). In training, we use random cropping for volumes, resulting in dimensions of $96 \times 96 \times 96$. During inference, a stride of $16 \times 16 \times 16$ is employed for efficient data processing.

### 3.2. Results

Table 1 illustrates the performance comparison between our proposed approach and the latest State-of-the-Art (SOTA) methods. Our method exhibits substantial enhancements across all metrics, excelling particularly in organ voxel detection, notably in Dice and Jaccard indices. This demonstrates that the incorporation of the uncertainty map along with the deep contrastive supervision can significantly enhance the efficacy of the model. In comparison to MCF, our technique is clearly superior, achieving a notable boost in DSC from 88.71 to 89.21 and a remarkable enhancement in the Jaccard index from 80.41 to 81.64, while simultaneously maintaining a superior 95HD metric. Furthermore, our method not only maintains a stable and reliable performance by minimizing variance but also excels in the visual comparison presented in Figure 3, showcasing its superiority in left atrial segmentation. The visuals emphasize the increased alignment with ground truth labels and a noticeable reduction in false segmentations, signifying the nuanced details effectively captured by our innovative approach.

Additionally, this robust performance also extends to the NIH Pancreas Dataset, as evidenced in the comprehensive results provided in Table 2. As the pancreas is situated deep within the abdomen, it exhibits notable variations in size, location, and shape. Adding to the complexity, pancreatic CT volumes present a more intricate background compared to the relatively simpler background of left atrial MRI volumes. This inherent complexity makes pancreas segmentation a more challenging task than left atrial segmentation. More specifically, our method outperformed all SOTA methods across all performance metrics in

Table 1: Comparison of results using the LA dataset (average ± standard deviation).

| Method | Dice(%)↑ | Jaccard(%)↑ | 95HD(voxel)↓ | ASD(voxel)↓ |
|---|---|---|---|---|
| MT (Tarvainen and Valpola, 2017) | 85.89 ± 0.024 | 76.58 ± 0.027 | 12.63 ± 5.741 | 3.44 ± 1.382 |
| UA-MT (Yu et al., 2019) | 85.98 ± 0.014 | 76.65 ± 0.017 | 9.86 ± 2.707 | 2.68 ± 0.776 |
| SASSNet (Li et al., 2020) | 86.21 ± 0.023 | 77.15 ± 0.024 | 9.80 ± 1.842 | 2.68 ± 0.416 |
| DTC (Luo et al., 2021) | 86.36 ± 0.023 | 77.25 ± 0.020 | 9.02 ± 1.015 | 2.40 ± 0.223 |
| MC-Net (Wu et al., 2021) | 87.65 ± 0.011 | 78.63 ± 0.013 | 9.70 ± 2.361 | 3.01 ± 0.700 |
| UCMT (Shen et al., 2023) | 88.13 ± 0.000 | 79.18 ± 0.000 | 9.14 ± 0.000 | 3.06 ± 0.000 |
| MCF (Wang et al., 2023) | 88.71 ± 0.018 | 80.41 ± 0.022 | 6.32 ± 0.800 | **1.90 ± 0.187** |
| **Our Method** | **89.21 ± 00.18** | **81.64 ± 00.24** | **6.31 ± 0.842** | 1.92 ± 0.195 |

pancreas segmentation. Expanding on the segmentation outcomes, Figure 3 offers deeper insights, emphasizing the consequential influence of the proposed modules on elevating the overall segmentation quality. Notably, our method excels by producing sharper edges and achieving more precise boundary separation compared to the MCF and MC-Net methodologies. This emphasizes its effectiveness in enhancing the reliability of object boundary predictions and distinctly discerning the organ of interest from the background.

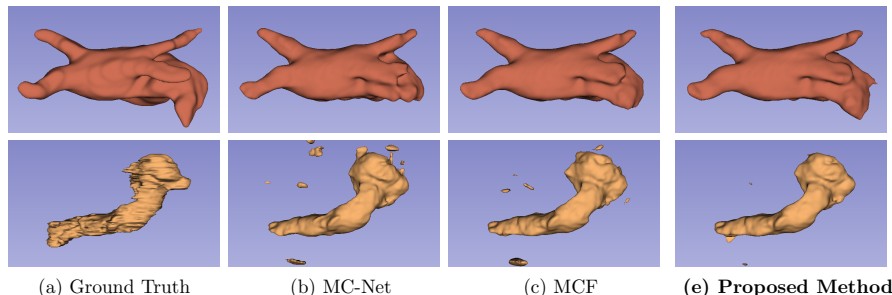

(a) Ground Truth    (b) MC-Net    (c) MCF    **(e) Proposed Method**

Figure 3: Visual comparison of segmentation results: the first and the second rows show the left atrium (LA) and pancreas, respectively.

Table 2: Comparison of results using the NIH dataset (average ± standard deviation).

| Method | Dice(%)↑ | Jaccard(%)↑ | 95HD(voxel)↓ | ASD(voxel)↓ |
|---|---|---|---|---|
| MT (Tarvainen and Valpola, 2017) | 74.43 ± 0.024 | 60.53 ± 0.030 | 14.93 ± 2.000 | 4.61 ± 0.929 |
| UA-MT (Yu et al., 2019) | 74.01 ± 0.029 | 60.00 ± 3.031 | 17.00 ± 3.031 | 5.19 ± 1.267 |
| SASSNet (Li et al., 2020) | 73.57 ± 0.017 | 59.71 ± 0.020 | 13.87 ± 1.079 | 3.53 ± 1.416 |
| DTC (Luo et al., 2021) | 73.23 ± 0.024 | 59.18 ± 0.027 | 13.20 ± 2.241 | 3.81 ± 0.953 |
| MC-Net (Wu et al., 2021) | 73.73 ± 0.019 | 59.19 ± 0.021 | 13.65 ± 3.902 | 3.92 ± 1.055 |
| MCF (Wang et al., 2023) | 75.00 ± 0.026 | 61.27 ± 0.030 | 11.59 ± 1.611 | 3.27 ± 0.919 |
| **Our Method** | **76.20 ± 0.022** | **62.33 ± 0.028** | **11.55 ± 2.703** | **3.10 ± 0.0980** |

## 4. Conclusion

In summary, our approach tackles challenges in semi-supervised medical image segmentation through the integration of two subnetworks aimed at identifying and refining uncertain predictions. The model employs both an uncertainty attention map and an uncertainty-aware descriptor to enhance accuracy in pixel-level segmentation, particularly in scenarios with inherent prediction uncertainty. The efficiency of our method is substantiated by results obtained from both left atrial and pancreas datasets.

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

## Appendix A. Ablation Study on the Effect of Suggested Modules

In our strategic approach, we incorporated both the deep contrastive learning algorithm and the MultiCURE Module to mitigate prediction uncertainty. To assess the impact of this loss function in reducing uncertainty, we have illustrated the uncertainty map of the network predictions on two sample data in Figure 4. Specifically, we conducted training with two settings: the first setting excludes the MultiCURE Module and deep contrastive learning, while the second setting includes them to address uncertainty.

Figure 4 indicates that, during the inference process, the network tends to produce lower confidence scores in cases without utilizing the uncertainty aware modules. However, including the MultiCURE Module and deep contrastive modules leads to a more substantial increase in the models' prediction confidence. This observation suggests that incorporating these modules enhances the network's confidence in predicting uncertain voxels, thereby improving overall prediction performance. Additionally, Table 3 has been provided to highlight the individual effect of each module on overall performance. Removing the MultiCURE Module results in a notable decline in model performance, and similarly, omitting deep contrastive learning leads to a slight drop in model performance.

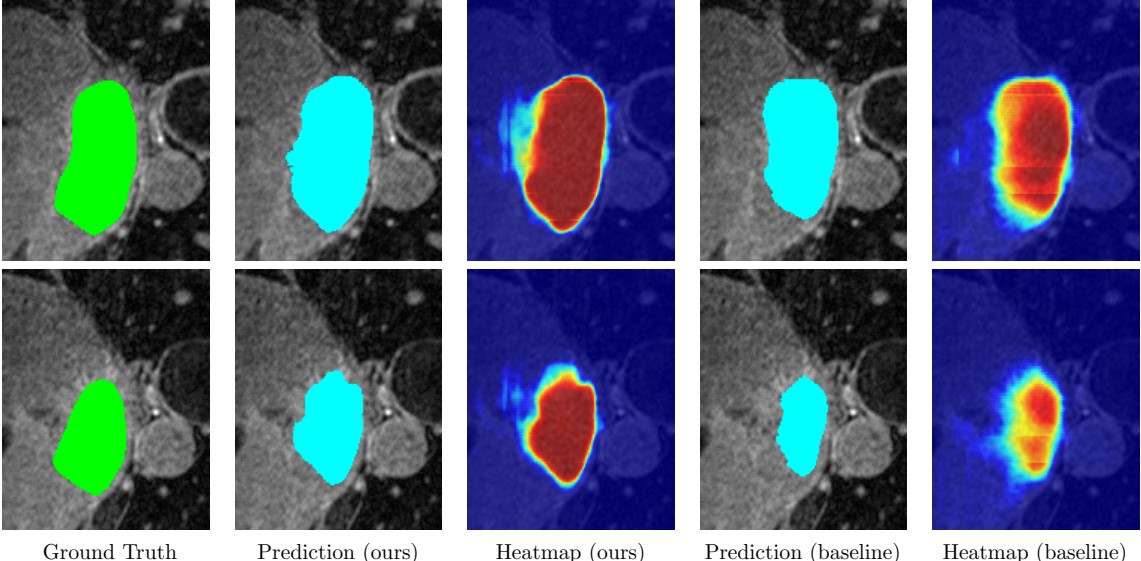

| Ground Truth | Prediction (ours) | Heatmap (ours) | Prediction (baseline) | Heatmap (baseline) |

Figure 4: Visualization of prediction maps and activation maps for two samples from the LA dataset using our suggested modules alongside the baseline model without the proposed enhancements.

The module's architecture allows the network to adapt its focus based on the characteristics of the input, providing a mechanism to selectively reduce uncertainty in regions where it is most crucial. This adaptability contributes to the observed improvement in the model's confidence, especially in predicting uncertain voxels. The results suggest that the MultiCURE Module enhances the network's ability to handle contextual uncertainties, leading to more confident and accurate predictions. Also, the contrastive loss is employed to guide the learning process. The results in Table 3 reveal that the absence of contrastive

Table 3: Effect of each suggested module on the overall performance using the Pancrese Dataset (average).

| $\mathcal{L}_{\text{contrastive}}$ | MultiCURE | Dice(%)↑ | Jaccard(%)↑ | 95HD(voxel)↓ | ASD(voxel)↓ |
|:---:|:---:|:---:|:---:|:---:|:---:|
| × | × | 74.10 | 61.00 | 12.30 | 4.12 |
| √ | × | 75.00 | 61.22 | 11.95 | 3.27 |
| × | √ | 75.80 | 62.33 | 11.58 | 3.25 |
| √ | √ | 76.20 | 62.33 | 11.55 | 3.10 |

learning negatively impacts the model's performance, emphasizing its role in enhancing feature representations. The contrastive learning mechanism enables the model to focus on relevant information, thereby improving its ability to make accurate predictions. The combination of the MultiCURE Module and contrastive learning synergistically contributes to the overall success of the proposed architecture. While the MultiCURE Module addresses contextual uncertainties, contrastive learning complements this by refining feature representations. The experimental results underscore the efficacy of this combined approach in achieving more robust and confident predictions, particularly in scenarios involving uncertain or ambiguous image data.

## Appendix B. Limitation

In our strategy, we proposed various mechanisms to increase the certainty of pseudo-labels, thereby enhancing the utilization of unlabeled data. While our method is designed to recalibrate features to improve voxel representation for certain predictions, uncertainty in voxel representation along object boundaries is often inherent due to abnormalities in the imaging device, making it challenging to distinguish between object boundaries and overlapped backgrounds. Despite our efforts, our method still faces limitations in enhancing prediction certainty along boundary regions, which are influenced by the characteristics of the imaging device. As illustrated in Figure 4, the predictions of the LA boundaries lack high confidence. It's noteworthy that this limitation is also encountered by expert radiologists in precisely distinguishing boundary voxels, especially in 3D volumes, where multiple raters are often employed to mitigate boundary errors by prioritizing the most agreed-upon regions (Ji et al., 2021).

Additionally, While our strategy focuses on enhancing the network receptive field size, incorporating our MultiCURE module, it is essential to acknowledge its limitations, particularly in scenarios involving small lesions, such as microbleedings in brain MRI. Despite our efforts to extend the receptive field size for capturing long-range dependencies, our approach may not adequately address the need for strong texture representation required for precise detection of such lesions, which involves both fine-grained and coarse features. This limitation highlights the challenges inherent in balancing the requirements of capturing both fine-grained and coarse features within a single model framework. Future research directions could explore specialized architectures or fusion techniques aimed at improving the detection sensitivity for these types of lesions.

