# OpenReview forum: "Reducing Uncertainty in 3D Medical Image Segmentation under Limited Annotations through Contrastive Learning"
_MIDL.io/2024/Conference — MIDL 2024 Poster_

### Official Review · Reviewer_8jsk · 2024-02-26

**Confidence:** 4
**Preliminary Rating:** 4
**Recommendation:** Poster

**Summary:**

This paper proposes a method to reduce uncertainty and improve 3D medical image segmentation through contrastive learning. They introduce two contributions: (i) an uncertainty-aware feature descriptor module that aims to improve contextual and semantic representations, and (ii) a contrastive learning function employed to reduce the differences between class prototypes and uncertain predictions. Their method is evaluated using k-fold cross validation in two tasks: left atrial and pancreas segmentation. They contrast their approach with various baseline methods and demonstrate a slight enhancement achieved.

**Strengths:**

- Method is properly explained and illustrated:
    - Uncertainty descriptor to refine the representation of erroneous predictions.
    - Class prototypes are obtained based on reliable predictions to build a deep contrastive supervision function.
- Evaluation with k-fold validation and comparison with several baselines.
- Ablation study to validate the contribution of the individual parts of the proposed method in Appendix.

**Weaknesses:**

- Choice of datasets is not justified.
- Limitations of the method are not discussed.
- No code available. The availability of the code would strength the transparency and reproducibility of the results.

**Detailed Comments:**

Paper is clearly written and well structured.

**Justification Of The Preliminary Rating:**

This paper is well motivated, and the method is extensively evaluated. It is compared against several baselines using k-fold validation. However, code is not available. The availability of the code would strength the transparency and reproducibility of the results.

**Questions To Address In The Rebuttal:**

1. Justify the choice of datasets. Is this method appropriate for any 3D medical image segmentation? How would it behave to segment small lesions?
2. Comment on the availability of the code.
3. Discuss foreseen limitations and ideas for future work.

**Special Issue:**

No

---

> ### Author Response · Authors · 2024-03-16
> **Response**
>
> #### 1. Justify the choice of datasets. Is this method appropriate for any 3D medical image segmentation? How would it behave to segment small lesions?
> > Thank you for bringing up this important aspect. Our method is intended to be a general approach applicable to a wide array of 3D medical imaging datasets, rather than being tailored to specific ones. Our experiments on both MRI (The LA dataset) and CT (Pancreas) volumes serve to demonstrate the effectiveness of our approach across different imaging modalities. These datasets represent challenging organ segmentation tasks in diverse anatomical regions and imaging modalities. By evaluating our method on such diverse datasets, we aimed to showcase its versatility and effectiveness across various modalities and challenging segmentation scenarios. It's worth noting that our method leverages convolutional layers with adjustable receptive field sizes, allowing for accurate localization of objects ranging from small to large sizes.
>
> #### 2. Comment on the availability of the code.
> > We have now included a link to the publicly available GitHub repository hosting our code.
>
> #### 3. Discuss foreseen limitations and ideas for future work.
> > We have included a new section "Limitation" in the appendix to address the limitations of our proposed method.

---

> ### Comment · Reviewer_8jsk · 2024-03-26
>
> Thanks to the authors for the revised version.
>
> While I understand that experiments on the LA dataset (MRI) and pancreas CT reflect different scenarios, this doesn't address my comment about small lesions (e.g. microbleedings in brain MRI) even when using convolutional neural networks. This could also be mentioned under limitations, if appropriate.

---

> > ### Author Response · Authors · 2024-03-27
> > **respond**
> >
> > #### While I understand that experiments on the LA dataset (MRI) and pancreas CT reflect different scenarios, this doesn't address my comment about small lesions (e.g. microbleedings in brain MRI) even when using convolutional neural networks. This could also be mentioned under limitations, if appropriate.
> > > many thanks for the feedback. In our strategy, we have focused on enhancing the network receptive field size, incorporating our MultiCURE module. This approach indeed extends the receptive field size, particularly beneficial for detecting organs requiring capturing long-range dependencies.
> > However, it's crucial to acknowledge that while this strategy proves effective for organs with extensive dependencies, it may encounter limitations in scenarios involving tiny lesions, such as microbleedings in brain MRI. These situations demand strong texture representation, which our current approach might not adequately address. o address this concern, we have revised the limitation section of the paper (see appendix) to underscore this aspect.

---

> > > ### Comment · Reviewer_8jsk · 2024-03-27
> > >
> > > Thanks for the explanation and the corresponding changes. I do not have further questions.

---

### Official Review · Reviewer_FbFC · 2024-02-26

**Confidence:** 4
**Preliminary Rating:** 2
**Final Rating:** 4

**Summary:**

In their paper, the authors tackle semi-supervised learning for 3-D medical data using convolutional neural networks, focussing on improving the representation within the network with different components. First, they propose to use uncertainty-maps combined with a feature descriptor to provide better contextualization especially in the case of uncertain samples. Second, they utilize a contrastive loss function for multi-layer deep supervision within the network to better align the data along class prototypes. They report results on two different 3-D datasets (MRI, left ventricle segmentation and CT, pancreas segmentation) and report improved results compared to the state-of-the-art.

**Strengths:**

- The paper is off to a good start, I enjoyed reading the introduction, which (for a short conference paper) was quite exhaustive.
- Interesting approach of focussing on uncertainty correction during model training using different approaches.
- Evaluation on two (small-mid sized) 3-D datasets, which presents an interesting use-case since annotation for 3-D images is especially time-consuming.

**Weaknesses:**

- Parts of the methodology is not clear to me, both from the point of motivation and from the way this is described (see details also below). Specifically I am missing a clear motivation of the need and the intricacies of the "MultiCURE" module and the benefit compared to a "simple" attention mechanism.
- The mathematical notation and wording does not seem to be fully consistent, with multiple synonyms being used.
- Some comparisons seems to be missing, like the - at least in the introduction prominently mentioned - approach by Shen et al. 2023.
- The experimental setup is not clear: How was the data split between supervised and unsupervised data parts? Did the authors follow Wang et al. in their setup? Did they use the same data splits?

**Detailed Comments:**

*Introduction*
- p. 3 "However, the issue of enhancing model representation and rectifying feature representation in uncertain regions remains unresolved.": Given the extensive introduction, the authors may want to spend an additional sentence on why this is still an issue with Shen et al.
- After having read the paper: It seems that Shen et al. is method-wise a close paper: Why is this approach not included in the comparison?

*Regarding the description of the proposed method - General setup:*
- p. 4: "we apply the unsupervised loss to the predictive model" - what exactly is here the unsupervised loss? Also, my feeling is that for Eq. 1 & 2, the notation could be clearer by directly defining the output of the predictive model and the output of the auxiliary model instead of only implicitly by "pseudo ground truth". In general, this part of the paper could be more clear, i.e.,
  - "subsequently utilize the uncertainty map to adjust the predictions of the second network" - it has not been defined what constitutes the "uncertainty map" or what the "second network" is
  - "both networks but with high confidence" - what do you mean to say here?
- Fig. 1: I'd suggest the authors review the draft for a consistent naming convention, e.g., they do not use any of the previous terms but now introduce f_A and f_B, L_R has not been presented before; additionally, it is not clear to me where the different representations (blue/red) for the contrastive loss come from.

*Regarding the description of the proposed method - MultiCURE:*
- p. 4: I don't fully understand what the authors mean to say here: "In our design, we argue that false predictions often result from the network’s uncertainty, and this can be rectified by adaptively adjusting the context representation." - false predictions in general? If I take this verbally, the authors should be able to show that the network is more uncertain for wrong predictions that for correct predictions. Is this the case?
- p. 4: "we use the first network’s predictions to create an uncertainty map, identifying low-confidence areas" - again, it is not clear to me how this uncertainty map is derived (directly from sigmoid/softmax? thresholded? different?) or rather how it is used then for the next step.
- p. 4/5: "MultiCURE, [...], is designed to recalibrate context information across multiple scales..." - This sentence and the following are not clear to me. Why recalibrate? What do the authors mean by "streamlines"? If the authors use 3x3 - 5x5 convs, why would this be global context?
- p. 5: "To further enhance feature representation within uncertain regions, we integrate an uncertain attention map with the deformable convolution output. This incorporation conditions the representation based on uncertain regions, treating it as a mechanism to accentuate the representation of uncertain regions using non-fixed resampling points." Again, this part is not really clear to me. How does this integration work? Is the uncertainty map multiplied with the output of the deformable convolution? If I understand the use of the deformable convolution correctly, it simply learns one offset per conv-filter [1], so the filter is no longer grid-aligned but still fixed.
- p. 6: Eq. (5)/Eq. (6): What is S_k? I presume the prototypes are generated from supervised samples only? What is the additional term that is incorporated? Furthermore, what format do c_k and v_k have? Since the spatial resolution decreases, how do the authors extract feature vectors sufficiently localized across different scales for a sensible contrastive loss function?
- p.6: "first and second path" - consider consistent notation

*Regarding the experiments:*
- p. 6: contrastive learning is reported to be unstable for small datasets. Did this cause any issues during training?
- p. 7/8: Information on data split and evaluation is completely missing (I presume the same setting as in Wang et al. 2023 was used and their results are simply re-printed here, but this is not clear from the paper description)
- p. 7/8: No fully / subset-supervised baselines with matching network reported

*Very minor:*
- consider using ".\ " in LaTeX to avoid very large spaces after non-sentence-ending periods, e.g., "et al.\ "
- p. 3: I'd use normal 1) / 2) / 3) instead of using ➊, etc.
- p. 5: "using offset field" - +  an
- Running title goes beyond the page (e.g., top of page 3)
- The case numbers for the left ventricle dataset seems to be wrong

*Additional comments, the authors may want to consider, also for future work*:
- I found the idea to pay more attention to uncertainty in the context of knowledge distillation quite interesting. Here, one very interesting on the topic, especially for the challenging and less frequent classes could also be to think about this also from a sub-type or bias aspect - how is this emphasized in semi-supervised learning and (how) can these biases potentially be better avoided with the proposed approach?

*References*:
- [1] https://arxiv.org/abs/1703.06211

**Justification Of Final Rating:**

I again thank the authors for substantially clarifying multiple aspects in their rebuttal and in the revised version of the paper. The approach presents an interesting approach to force the network to pay more attention to uncertain regions in the image. While I would have liked to see a broader evaluation with different attention-based approaches, I raise my rating to a weak accept.

**Justification Of The Preliminary Rating:**

The paper is off to a good start; however, continuing the paper, I have the feeling that the information is presented less clear and less complete. As such, the details of the methodology remained unclear to me and information about how the approach was evaluated is missing. From my perspective and at this point, the paper requires a considerable revision to clarify these aspects as well as potentially additional comparisons that go beyond the ablation study presented in the appendix.

**Questions To Address In The Rebuttal:**

I'd like the authors to answer the following questions:
- what is the benefit of using the MultiCURE compared to an attention mechanism?
- clear out potential misunderstandings with regard to the underlying workings of the proposed method, especially with regard to the uncertainty map and the computation of the contrastive loss
- how exactly was the data employed for semi-supervised learning? How was train and test data handled?

---

> ### Author Response · Authors · 2024-03-16
> **Responses provided in order. Many thanks for all the comments.**
>
> #### p. 3 "Ho ..
> > We have revised the introduction section to further emphasize the limitation of the Shen et al. method in addressing the need for rectifying feature representations in uncertain regions.
> #### After .?
> > We have incorporated the results of the Shen approach on the LA dataset into our manuscript. Upon comparison, it is evident that our method outperforms Shen's approach.
> #### p. 4: "we ..?
> > We have revised the equations accordingly and made the passage clearer for the reader. Specifically, we have introduced the terms "predictive" and "auxiliary" through the definitions to make it explicit.
> #### Fig. 1..
> > We have updated Figure 1 to ensure consistency with the text's notation throughout the manuscript. In the revised figure, the red and blue points now represent voxel representations belonging to the foreground and background classes, respectively. These representations are employed by the contrastive loss to promote the model's learning of discriminative features between voxels of two classes (e.g. LA and background). This clarification enhances the understanding of how the contrastive loss functions within our proposed method.
> #### p. 4: I ..?
> > What we intended to convey was not that false predictions generally arise from uncertainty. Instead, we aimed to highlight that inadequate contextual information in regional representation can lead to inaccuracies. These regions often lack sufficient contextual cues, posing challenges for precise predictions. By adaptively adjusting the context representation in these uncertain regions, our method aims to alleviate this uncertainty, thus minimizing false predictions.
> #### p. 4: "we ..
> > We have updated the paper with additional details. Specifically, we employed the threshold strategy and considered predictions with a confidence level below 0.7 to be uncertain.
> #### p. 4/5: "Multi ..?
> > Recalibration involves strategically refining the network's focus on crucial features through the adaptive adjustment of the sizes of the receptive fields it examines. The deformable convolution in the third path is designed to dynamically adjust the sampling grid, enabling the model to adaptively capture information from larger spatial extents based on the characteristics of the input data. This approach, especially effective on lower-dimensional feature maps, contributes to a better understanding of the global context. Hence, this is why we refer to it as contributing to global understanding. "Streamlines" refers to multiple pathways within the MultiCURE module that process data concurrently. We replaced "streamlines" with "paths" in the manuscript for improved clarity and readability.
> #### p. 5: "To ..
> > The uncertain attention map specifically targets areas within the input data where the model exhibits low prediction confidence. By assigning certain regions as zero and preserving the details of uncertain regions, this map focuses on enhancing the representation of areas needing further attention. Integration with the output of a deformable convolution is achieved through a straightforward summation process. In the architecture's first two pathways, the focus is primarily on the regions deemed certain, where the model's predictions are more reliable. However, the path involving the uncertain attention map diverges by prioritizing information considered uncertain. This approach intentionally amplifies the features within these uncertain regions, directing the model toward a better understanding of these areas.
> #### p. 6: Eq ..?
> > We have carefully reviewed all the equations and ensured that all notations are properly defined throughout the document.
> #### p.6: "first ..
> > revised.
> #### p. 6: co ..
> > In our approach, we integrate the contrastive loss with a supervised loss function, which guides the contrastive loss toward better convergence. As a result, we did not encounter any significant fluctuations or divergence issues during the training process.
> #### p. 7/8: Info..
> > We have now clarified our evaluation setting regarding k-fold cross-validation.
> #### p. 7/8: No
> > Our baseline models, following the MCF (Wang et al.) strategy and using VNet and 3D-ResNet architectures, show low performance with subset-supervised learning, as presented in the MCF paper.
> #### Very minor
> > Addressed.
> #### what .?
> > MultiCURE aims to reduce prediction uncertainty through adaptive feature recalibration, especially in highly uncertain regions. Unlike standard attention mechanisms that enhance features based on relevance, we aim to extend attention not only to object regions but also to shift it to other areas where the model has difficulty classifying. MultiCURE adaptively selects kernels from two paths that focus on certain regions and one path from uncertain regions, all in a soft-attention manner. This approach offers adaptability to variations in organ shapes and ensures the precision required for boundary delineation.
> #### clear ..
> > see above.
> ####  how .?
> > see above.

---

> > ### Comment · Reviewer_FbFC · 2024-03-23
> > **Response to Author Comments**
> >
> > I thank the authors for providing a revised version.
> >
> > I would like to mention that the way that the form of the rebuttal makes it somewhat difficult to follow the argumentation as they structured their reply along the detailed comments and not along the main points to be answered in the rebuttal. It further requires to side by side check the original comments, identify the corresponding answer and the revised manuscript, since many answers are very short. While I understand that OpenReview has a max. character number per message, this is not a restriction set by MIDL, and I have seen this solved in other cases on Open Review by simply posting a two-part response.
> >
> > Generally, the revised version of the paper clarifies many of my previously raised questions.
> >
> > Some aspects the authors may want to further consider/clarify:
> > - *deformable convolutions*: the authors may want add a reference to the paper on Deformable Convolutional Networks if this is the approach that they have used.
> > - Global vs. local: Did the authors investigate the offsets predicted by the deformable convolution?
> > - Do I understand it correctly that the uncertainty map is a binary map? Do the authors then simply "add" a constant bias (of 1) for all locations the network is uncertain? I did not find a statement corresponding to "by assigning certain regions as zero ..." in the paper - could you kindly point me to it?

---

> > > ### Author Response · Authors · 2024-03-23
> > > **Response**
> > >
> > > We regret any inconvenience caused by the format of our rebuttal. We acknowledge your suggestion and commit to implementing it in our future rebuttals. Thank you once again for your comments. Below, we have provided our response and also updated the paper accordingly.
> > >
> > >
> > > #### deformable convolutions: the authors may want add a reference to the paper on Deformable Convolutional Networks if this is the approach that they have used.
> > > > Thank you very much for bringing up this point. We have now included the reference for the deformable convolution paper.
> > >
> > > #### Global vs. local: Did the authors investigate the offsets predicted by the deformable convolution?
> > >
> > > > In our experiments, we indeed replaced the deformable convolution with the regular convolution and observed a decrease in performance. Conversely, our method utilizing the deformable convolution consistently yielded superior performance. Additionally, we found that the deformable convolution tended to learn larger offsets compared to a 3x3 convolution, enabling better capture of global information. It's worth noting that the offset learned by the deformable convolution also depends on the characteristics of the data and the complexity of the patterns present within it. With this approach, we give freedom for the model to capture local to global information more effectively, rather than limiting the receptive field size.
> > >
> > > #### Do I understand it correctly that the uncertainty map is a binary map? Do the authors then simply "add" a constant bias (of 1) for all locations the network is uncertain? I did not find a statement corresponding to "by assigning certain regions as zero ..." in the paper - could you kindly point me to it?
> > > > Yes, you are correct. The uncertainty map is indeed a binary map. In our approach, we define uncertain regions as those where the network's confidence score falls below a specified threshold, assigning them a value of 1. Conversely, regions, where the confidence score exceeds the threshold, are labeled as certain and assigned a value of 0. We have updated section 2.1 to clarify this process.

---

### Official Review · Reviewer_WLFM · 2024-02-28

**Confidence:** 4
**Preliminary Rating:** 4
**Final Rating:** 4

**Summary:**

This paper proposes a semi-supervised contrastive learning approach using uncertainty, extending existing pseudo-labeling methods with confidence score filtering for semi-supervised training.
Two subnetworks are used: The first one identifies uncertain predictions, generating an uncertainty attention map. The second one employs an uncertainty-aware descriptor to refine the representation of uncertain regions, enhancing the accuracy of predictions.

**Strengths:**

The design of the proposed method is good and obtains good results on the two datasets used for evaluation. It compares well with the SoA. An ablation study also shows the benefit of the different components.

**Weaknesses:**

The method could be better motivated (more details below).

There is important missing information on the splits.

Why not using the 54 test cases of the challenge to be able to compare against the participants ? Results in the leaderboard seem higher.

No statistical test is performed.

**Detailed Comments:**

As a general comment, I would motivate better why trying so much to reduce uncertainty. Border voxels are naturally uncertain, as reflected by inter- and intra-observer variability. The related works motivate the need for reducing uncertainty of pseudo-labels. And your motivation "However, the issue of enhancing model representation and rectifying feature representation in uncertain regions remains unresolved." is not evident to me.

"consists of 1003 3D" -> 100, not 1003 !

"we employed k-fold cross-validation." Missing important information: what is k? Are the k-1 folds split into train/validation and with which percentage?
I assume it is the same splits as in (Wang et al, etc.), but you need to clarify this.

Clarify if you report standard deviation or standard error in Tables 1 and 2.
And report it also in Table 3.

Make sure to define all notations (eg S_k, Eq. 5)

Some typos, eg "from different level"

What is the link with you arxiv paper "Leveraging Unlabeled Data for 3D Medical Image Segmentation through Self-Supervised Contrastive Learning" ?
Why are these results not included/compared with?

**Justification Of Final Rating:**

The authors responded to most of my comments, except for those in the "Weaknesses" section (on the test cases of the challenge and the statistical tests). The overall quality of the paper was improved. My recommendation remains unchanged (weak accept).

**Justification Of The Preliminary Rating:**

Except for the general motivation of reducing uncertainty (in my comments), the approach is well motivated.
The method seems robust, with good results on the non-extensive experiments with the two public datasets.

**Questions To Address In The Rebuttal:**

My comments are rather minor and easy to address I think.
A better motivation for the main part of reducing uncertainty would help, and fixing minor errors in the document and adding some missing information.

---

> ### Author Response · Authors · 2024-03-16
>
> #### 1. As a general comment, I would motivate better why trying so much to reduce uncertainty. Border voxels are naturally uncertain, as reflected by inter- and intra-observer variability. The related works motivate the need for reducing uncertainty of pseudo-labels. And your motivation "However, the issue of enhancing model representation and rectifying feature representation in uncertain regions remains unresolved." is not evident to me.
> > Thank you for your insightful comment. We have carefully revised the introduction section to better present the motivation behind our proposed method, specifically addressing the limitations of existing literature.
>
> #### 2. "consists of 1003 3D" -> 100, not 1003 !
> > Thank you for pointing that out. We have corrected the number of samples accordingly.
>
> #### 3. "We employed k-fold cross-validation." Missing important information: what is k? Are the k-1 folds split into train/validation and with which percentage? I assume it is the same splits as in (Wang et al, etc.), but you need to clarify this.
> > Thank you for highlighting this important point. We have now clarified our evaluation setting regarding k-fold cross-validation. Specifically, we follow the methodology of Wang et al. [1], utilizing 5-fold and 4-fold cross-validation for the LA and Pancreas datasets, respectively. Each fold is split into train/validation sets with the same percentages as described in the [1] method.
>
> #### 4. Clarify if you report standard deviation or standard error in Tables 1 and 2. And report it also in Table 3.
> > Thank you for the feedback. We have revised the captions of the tables and emphasized this by including "(average ± standard deviation)."
>
> #### 5. Make sure to define all notations (eg S\_k, Eq. 5)
> > We have carefully reviewed all the equations and ensured that all notations are properly defined throughout the document.
>
> #### 6. Some typos, eg "from different level"
> > We thoroughly checked the manuscript to correct any typo errors.
>
> #### 7. What is the link with you arxiv paper "Leveraging Unlabeled Data for 3D Medical Image Segmentation through Self-Supervised Contrastive Learning" ? Why are these results not included/compared with?
> > Thank you for your inquiry. Our recent arXiv paper [2] proposes a distinct strategy for enhancing medical image segmentation on 3D medical images. As this paper has not been formally published yet, we have not directly compared it with our current approach. However, it's worth noting that our current method outperforms the approach proposed in [2] on the LA dataset and demonstrates comparable performance on the pancreas dataset. While we acknowledge the relevance of our arXiv paper, we focused our comparisons on published methods to ensure a comprehensive evaluation against existing state-of-the-art techniques.
>
> References:
> ###### [1] Zhiqiang Shen, Peng Cao, Hua Yang, Xiaoli Liu, Jinzhu Yang, and Osmar R Zaiane. Co-training with high-confidence pseudo labels for semi-supervised medical image segmentation. Proceedings of the Thirty-Second International Joint Conference on Artificial Intelligence, 2023.
> ###### [2] Karimijafarbigloo Sanaz, Azad Reza, Velichko Yury, Bagci Ulas, Merhof Dorit. Leveraging Unlabeled Data for 3D Medical Image Segmentation through Self-Supervised Contrastive Learning. arXiv:2311.12617, 2023.

---

> > ### Comment · Reviewer_WLFM · 2024-03-26
> >
> > Thank you for the response and the modifications.
> >
> > Two comments that were in the "Weaknesses" section were:
> >
> > Why not using the 54 test cases of the challenge to be able to compare against the participants ? Results in the leaderboard seem higher.
> >
> > No statistical test is performed.

---

> > > ### Author Response · Authors · 2024-03-27
> > > **respons**
> > >
> > > Thank you for your suggestion. In our approach, we have already provided the mean and variance of our method for each metric, following the exact setting as presented in the MCF paper. This ensures consistency and enables direct comparison with the results reported in the MCF paper. Additionally, it's important to note that while we have not tested our method on the challenge test set, we anticipate that our method, under the same setting, would also produce acceptable results. This anticipation stems from the fact that our method does not rely on any prior conditions or dependencies specific to the dataset.

---

### Meta-Review · Area_Chair_35Rc · 2024-04-02

**Recommendation:** Accept (Poster)
**Confidence:** 5

**Metareview:**

This work innovates in integrating/utilizing uncertainty properties within the Mean-Teacher paradigm. The to key components are the use of a dedicated attention module and the design of a new contrastive loss. The performance of the method was favorably evaluated with state of the art methods on two different datasets (segmentation of the Left atrium from MRI and segmentation of pancreas from CT images).

The strengths of this work are:
1) the two methodological innovations that allow uncertainty to be exploited in the Mean-Teacher formalism
2) the ablation study, which proves the usefulness of each component
3) the comparison of the methods with 7 state of the art methods

The weaknesses of this work are:
1) the lack of comparison of the attention module with more conventional attention strategies
2) the description of the contrastive loss which should be improved
3) the proportion of label and unlabel data used during the experiments is not clearly mentioned in the article

This study proposes interesting methodological innovations for the development of weakly supervised methods. The authors have responded positively to most of the comments made by the reviewers, which strongly improves the quality of their paper.

For all these reasons, I have decided to accept this article.

---

### Decision · Program_Chairs · 2024-04-06

Accept (Poster)